# A Critical Review of Risk Assessment Models for *Listeria monocytogenes* in Produce

**DOI:** 10.3390/foods13071111

**Published:** 2024-04-04

**Authors:** Ursula Gonzales-Barron, Vasco Cadavez, Juliana De Oliveira Mota, Laurent Guillier, Moez Sanaa

**Affiliations:** 1Centro de Investigação de Montanha (CIMO), Instituto Politécnico de Bragança, Campus de Santa Apolónia, 5300-253 Bragança, Portugal; vcadavez@ipb.pt; 2Laboratório para a Sustentabilidade e Tecnologia em Regiões de Montanha, Instituto Politécnico de Bragança, Campus de Santa Apolónia, 5300-253 Bragança, Portugal; 3Nutrition and Food Safety Department, World Health Organization, 1202 Geneva, Switzerland; 4Risk Assessment Department, French Agency for Food, Environmental and Occupational Health & Safety (Anses), 14 rue Pierre et Marie Curie Maisons-Alfort, 94700 Maisons-Alfort, France; laurent.guillier@anses.fr

**Keywords:** systematic review, exposure assessment, listeriosis, leafy greens, vegetables, fruits

## Abstract

A review of quantitative risk assessment (QRA) models of *Listeria monocytogenes* in produce was carried out, with the objective of appraising and contrasting the effectiveness of the control strategies placed along the food chains. Despite nine of the thirteen QRA models recovered being focused on fresh or RTE leafy greens, none of them represented important factors or sources of contamination in the primary production, such as the type of cultivation, water, fertilisers or irrigation method/practices. Cross-contamination at processing and during consumer’s handling was modelled using transfer rates, which were shown to moderately drive the final risk of listeriosis, therefore highlighting the importance of accurately representing the transfer coefficient parameters. Many QRA models coincided in the fact that temperature fluctuations at retail or temperature abuse at home were key factors contributing to increasing the risk of listeriosis. In addition to a primary module that could help assess current on-farm practices and potential control measures, future QRA models for minimally processed produce should also contain a refined sanitisation module able to estimate the effectiveness of various sanitisers as a function of type, concentration and exposure time. Finally, *L. monocytogenes* growth in the products down the supply chain should be estimated by using realistic time–temperature trajectories, and validated microbial kinetic parameters, both of them currently available in the literature.

## 1. Introduction

Invasive listeriosis is a rare but severe foodborne disease that affects certain population groups such as the elderly, pregnant women and neonates, and immunocompromised individuals (i.e., patients with diabetes, AIDS, cancer and inflammatory diseases) [1]. According to EFSA BIOHAZ [2], demographic changes and health status factors, as well as a preference switch of the consumers over high-risk ready-to-eat (RTE) foodstuffs would produce an increase in listeriosis incidence. Surveillance data have shown that the occurrence of *Listeria monocytogenes* in RTE products has been responsible for outbreaks worldwide. According to EU data [3], in the period between 2010 and 2020, a share of 10% of the total strong-evidence outbreaks of listeriosis was linked to produce, having raw vegetables, juices and black olives as sources of the disease. Unlike the EU scenario, in the USA, produce presented a higher share (30%) in the ten-year span, being the second topmost food category (following dairy) linked to strong-evidence listeriosis outbreaks [4]. In the USA, pre-cut celery, cantaloupe, frozen vegetables, peaches/nectarine, mung bean sprouts, pre-packed leafy greens, sprouts, pre-packed lettuce, enoki mushrooms, avocado, and stone fruit were the food vehicles associated with listeriosis outbreaks.

Many case–control studies of sporadic listeriosis in susceptible populations pointed out that vegetables could be significantly linked to listeriosis. Carrots, retail vegetables and RTE mixed salads presented odds ratios (OR) of 2.00 (95% CI: 0.90–4.54) in the perinatal and non-perinatal US population [5], 1.92 (95% CI: 1.32–2.78) in the perinatal and non-perinatal UK population [6], and 1.72 (95% CI: 1.20–2.47) in the UK non-perinatal UK population [7], respectively. In the case of fruits, a meta-analysis investigation [8] obtained a significant pooled OR of 1.54 (95% CI: 1.14–2.07) for the association between sporadic listeriosis and the consumption of watermelons, melons and cantaloupes by susceptible population in the USA [5] and Australia [9].

Various listeriosis quantitative risk assessment (QRA) models have been produced for RTE foods in general [10] in an attempt to assess the efficacy of current practices or potential strategies that retard or prevent the development of this pathogen. The objectives of this study are: (i) to accomplish a critical review of the published QRA models of listeriosis linked to RTE and non-RTE produce; (ii) to compare the relative effectiveness of the control measures or strategies evaluated in the various QRA models as simulated scenarios; and (iii) to elicit, thereof, recommendations for future QRA models in produce.

## 2. Materials and Methods

QRA models were retrieved through a literature search on Scopus and PubMed^®^ considering 1998 as the starting year of publication. The searches in title, keywords and abstract were performed on 18 May 2022, using logically connected terms ((“risk assessment” OR exposure OR quantitative microbial OR risk modelling OR modeling OR simulation* OR second-order OR “second order” OR “risk management”) AND (“*L. monocytogenes*” OR “*Listeria monocytogenes*” OR listeriosis)) properly structured according to the syntaxes of the literature search engines. The full systematic review process and extraction of information have been described in Gonzales-Barron et al. [7]. The present review focuses only on produce, which is the subject of 13 QRA models described in 12 publications [11,12,13,14,15,16,17,18,19,20,21,22].

## 3. Results

A total of 13 QRA models on produce as a source of listeriosis were recovered in the literature search of models published between January 1998 and May 2022. Table 1 compiles the main features of the 13 QRA models, whereas Table 2 summarises the predictive microbiology models used in their construction, and the key outcomes from what-if scenarios and sensitivity analysis.

Eight models (61.5%) evaluated RTE produce, including lettuce salad [12,13], fresh-cut romaine lettuce [14], leafy greens from salad bars [16,18], leafy vegetables [17], fruits and vegetables [19] and fresh-cut cantaloupe [14]; whereas, five models (38.5%) focused on non-RTE produce, namely, fresh lettuce [11,20], fresh baby spinach [15] and frozen vegetables [21,22].

Most of the available models (11/13) characterised the conditions of the USA [14,15,19,21] and European countries, namely Spain [12,20], France [13], and the Netherlands [16,18], in addition to a model using data from the EU [22]. The other two QRA models pertained to the risk of listeriosis evaluated in Korea [11] and Brazil [17] (Table 1).

The QRA models were variable in terms of the scope of the food chain they represented. Four models simulated the growth of *L. monocytogenes* from processing to table [12,13,14,15], three models from end of processing to table [16,17,18], two models from retail to table [19,20], and two models represented only the consumption module [21,22]. Only one QRA model evaluated the contamination from production to consumption [11], although the actual utility of the model for the assessment of control measures is not clear since they did not assess risk factors or intervention strategies (Table 1).

The construction of models varied in complexity. Five out of the 13 QRA models mimicked cross-contamination taking place at processing (during cutting [14], before packaging [15], and during packaging [14]), during transport and retail [11], and during consumer’s handling [11,20]. All the QRA models relied on predictive microbiology models to estimate the concentrations of *L. monocytogenes* as the raw materials (i.e., vegetables or fruits) went through processing, and as end products went down the supply chain. The most widely used primary models for growth were the loglinear [12,17,18,19,20,21] and the Baranyi and Robert’s model [14,15,16], whereas the square-root model was the preferred choice of secondary model in all cases [11,12,14,15,16,17,18,19,20,21], except that of Crèpet [13] where the cardinal parameter model for temperature was employed (Table 2). The lag phase duration of *L. monocytogenes* in produce was taken into account in six models [11,13,14,15,21]; while only three QRA models employed time–temperature profiles to estimate the growth of *L. monocytogenes* during cold storage, including display at deli salads establishments [13,16,18] (Table 1). Crépet [13] explored the impact of maximum bacterial levels. The comparison of predicted and observed counts of naturally contaminating *L. monocytogenes* in minimally processed fresh lettuce showed inconsistency. The maximum levels observed in challenge tests were higher than those observed in naturally contaminated products. The maximum level during the stationary phase was better predicted by considering the initial contamination [30]. All the 13 QRA models regarded illness as the endpoint for risk estimation. The most commonly chosen dose–response function for risk characterisation was the exponential function, although in different approaches, namely the exponential model of FAO-WHO [23] (used in [11,13,20]), the one of Chen et al. [25] (used in [15,16,18]), the one of FDA-FSIS [19] (used in their own QRA model), the one of Buchanan et al. [26] (used in [17,21]), and the one of EFSA-BIOHAZ [2], based on the model of Pouillot et al. [29] (used in [22]).

The early Weibull-gamma model proposed by Farber et al. [24] was used in the three QRA models [12,14] and the dose–response of Bemrah et al. [28] used in the QRA model of Zoellner et al. [21]. The only QRA model that, in addition to illness, considered death as endpoint for risk estimation was that of Tromp et al. [18], which used a dose–response animal model in pregnant guinea pigs [27] (Table 1).

In total, 10 out of 13 QRA models assessed the effect of what-if scenarios on the final risk [12,13,14,15,16,17,18,21,22], whereas sensitivity analysis on response variables such as *L. monocytogenes* concentration at consumption, risk per serving or number of listeriosis cases was undertaken in 7 QRA models [12,13,16,18,20,21,22] (Table 2).

Although the outcomes of the QRA models are not directly comparable due to differences in model architecture, data and assumptions, taking together the results from the what-if scenarios and sensitivity analysis, the factors pointed out as having the biggest influence on reducing the likelihood of illness in the consumers are (Table 2): lower initial contamination and prevalence of *L. monocytogenes* in fresh vegetables, processing interventions such as ionising radiation or cold-atmospheric plasma, modified atmosphere packaging, reducing cross-contamination during processing, washing vegetables with chemical sanitisers, the maintenance of low storage temperatures, within-lot microbiological testing of the product at the end of processing, and a reduction in shelf-life. These factors are discussed in depth in the next section.

## 4. Discussion

The fact that most of the listeriosis QRA models were undertaken for leafy greens (9/13: as fresh, RTE and in salad bars) is evidence of the researchers’ concern from a microbiological safety perspective. Most leafy greens have a short growing season; therefore, they may be more vulnerable to microbial contamination. They are commonly grown outdoors, where they may be exposed to contaminated soil, fertilisers and irrigation water. In addition, produce such as leafy greens are mostly consumed raw. When minimally processed, the processing steps of washing, sanitising and modified atmosphere packaging do not guarantee the elimination of *L. monocytogenes*. Leafy greens and other produce have therefore far fewer barriers against *L. monocytogenes* than the more traditional vehicles of foodborne illness (i.e., cheese, RTE meat products).

### 4.1. Risk Factors on Farm

*L. monocytogenes* is ubiquitous in natural environments; it has been isolated from soil, water ways, vegetation, and from the faeces of domestic and wild animals [31]. Such environments are associated with the production of produce, and hence, the pathogen can be transferred from multiple transmission routes such as air, soil, water, insects, animals and human activity [32]. Furthermore, production activities including irrigation, fertilisation, and other on-farm management practices can affect *L. monocytogenes* prevalence on farm.

The type of cultivation system is the first on-farm risk factor that should be taken into account, because from a microbiological perspective, protected cultivation (i.e., greenhouse) is considered safer than open field, which is related to the minimisation of the risk factors linked with the sources of pre-harvest contamination, and a greater control of water disinfection with water being recirculated and cleaned periodically [33]. Another major risk factor, particularly in the contamination of leafy crops eaten raw as salads, is the agricultural water [34]. The main irrigation water sources are municipal water, rainwater, groundwater and surface water. The prevalence of *L. monocytogenes* in irrigation water appears to be variable; yet, surface water represents one of the riskiest water sources [35]. Irrigation methods also influence the microbiological safety of fresh produce. Song et al. [36] found out that furrow irrigation resulted in greater microbial contamination of lettuce and soil surface than subsurface drip irrigation. Settanni et al. [37] demonstrated that watering pots containing soil artificially contaminated with *L. monocytogenes* from sub-irrigation did not determine the contamination of the aerial parts of leafy vegetables, and, for this reason, the transmission of pathogens by drip irrigation is very low in comparison to the irrigation performed by overhead sprinklers. Even if direct contact between irrigation water and the edible part of the leafy crop is avoided, irrigation water may contaminate the soil, where the bacteria can survive for some time, and irrigation (or rainfall) splash may contaminate the crop [35]. Furthermore, fields that contain animal manure are more likely to be contaminated with *L. monocytogenes*, as shown by Szymczak et al. [38], because of their ability to survive in soils for months. Inadequate composting is also responsible for the transmission of undesired microorganisms from manures to the soil [33]. Strawn et al. [39] and Weller et al. [40] concluded that good manure management practices, such as aging, treating, and handling before application, are essential, since they can significantly influence the risk of *L. monocytogenes* contamination.

Despite the broad understanding that the microbial contamination of produce begins at the pre-harvest stage, none of the QRA models retrieved has looked into pre-harvest factors as sources of *L. monocytogenes* on farm. The only on-farm strategy tested among all available QRA models was the application of a microbiological criterion at primary production, consisting of *n* = 20, c = 0 and absence in 25 g. In the model of Carrasco et al. [12], such intervention would reduce the mean cases of listeriosis due to consumption of RTE lettuce salads by 43%. Carrasco et al. [12] argued that their QRA model covered the scope from processing to table, since on-farm factors influencing the status of *L. monocytogenes* in vegetables are challenging to control. Nonetheless, from a full supply chain model, a deeper understanding can be gained on the contributions to risk from important on-farm sources such as type of cultivation, water, fertilisers and irrigation. According to Olaimat and Holley [41], the pre-harvest microbial contamination is of utmost importance in keeping up the microbiological quality of fresh and minimally processed vegetables, because the post-harvest sanitisation applied during processing does not pursue the complete elimination of pathogens on plant surfaces. In fact, the sanitising washing procedure achieves a limited reduction in spoilage and pathogenic microorganisms. Instead, the application of on-farm safety procedures can reduce the burden of *L. monocytogenes* at the processing stage. Therefore, the construction of a QRA model that encompasses an on-farm module is needed, as it can help assess the effects of current on-farm management practices and preventive measures as well the effects of the type of cultivation on the final risk to the consumers associated with fresh and minimally processed produce. Now that dose–response relationships based on strain genetics have been proposed [42,43], it would be important to look at the genetic diversity of *L. monocytogenes* in agricultural soils. Clonal complexes (CC) 1, CC2, CC4, and CC6 are infection-associated clones usually causing sporadic or outbreak listeriosis, while CC9 and CC121 are strongly food-associated clones that mostly infect immunocompromised individuals [44].

### 4.2. Risk Factors at Processing

*L. monocytogenes* may enter the processing facilities through contaminated produce and personnel. Once introduced in the environment, the pathogen can grow at operational temperatures and resist cleaning and disinfection of the processing plant, given its ability to persist on abiotic surfaces such as stainless steel and polystyrene. Furthermore, *L. monocytogenes* can survive on the surface of fresh produce for extended periods of time [45], whereas contamination on an injured leaf may lead to growth and colonisation [46].

Except for irradiation after packaging, there is no step that can be highly lethal against *L. monocytogenes* in the processing of RTE produce. Therefore, the microbiological safety of leafy greens should rely on sanitising, washing and the maintenance of cold temperatures, in addition to good management practices and good hygiene practices.

As opposed to the void observed among the QRA models retrieved for an on-farm module, five QRA models simulated processing modules [12,13,14,15] and assessed the effects of a range of processing stages or interventions, namely, ionising radiation, cold atmospheric plasma (CAP), sanitisation and modified atmosphere packaging (MAP). Among the strategies tested by Guzel [14] and Omac et al. [15], irradiation appeared as the most effective intervention to reduce the exposure and the risk of listeriosis. In the QRA model for RTE fresh-cut romaine lettuce, exposure to ionising radiation of 1 kGy at room temperature decreased *L. monocytogenes* counts at the point of consumption by >99% and risk of illness in 1.66 log in the susceptible population, whereas in a QRA model for fresh cantaloupe by the same authors, the implementation of irradiation also reduced *L. monocytogenes* counts at the point of consumption by >99.9% and the mean cases of listeriosis by 99%, in comparison to baseline scenarios without an irradiation process [14]. In a QRA model for non-RTE baby spinach, Omac et al. [15] compared scenarios of water washing and washing plus irradiation with a baseline scenario representing neither interventions nor cross-contamination during processing. They found that irradiation was far more effective than water washing since washing decreased mean cases of listeriosis by 7.5% whereas adding irradiation to this scenario decreased mean cases of listeriosis by 56%, both in comparison to the baseline scenario (Table 2). Although the irradiation technology has been proven to be more effective than mild technologies [14,15], many consumers are still reluctant to accept irradiated foods, therefore making its adoption by the industry difficult. Crépet [13] assessed the impact of the chlorination of water. It was found that removing the treatment of water (baseline scenario) increased the risk by two.

Mild technologies of produce preservation such as CAP and MAP were also assessed in QRA models; where the use of MAP (5.5% CO_2_, 3% O_2_; 92.5% N_2_) as opposed to no packaging (baseline) reduced the mean number of listeriosis cases associated with RTE lettuce salad by 95% [12], treatment of romaine lettuce with CAP eliminated on average 92% of the population of *L. monocytogenes* at the point of consumption, reducing in 1.34 log units the risk of illness in the susceptible population [14].

Mild decontamination treatments such as washing and sanitising with chemicals take place during processing. Such a decontamination step plays an important role in the reduction or prevention of pathogenic contamination in fresh produce since the processing of produce does not include any critical control point such as heat treatment, sterilisation or freezing to control the microbial load. Eco-friendly sanitisers, as opposed to the widely known chlorine which forms by-products that may cause negative effects on human health and environment (i.e., trihalomethanes, halo acetic acids and haloketones), are being applied in fresh cut industry, including chlorine dioxide (ClO_2_), peroxyacetic acid (PAA) and electrolysed water (EW) [47]. In his QRA model, Guzel [14] estimated that, when no cross-contamination during processing was considered, the use of PAA for sanitising reduced *L. monocytogenes* mean concentration at consumption by 28% and reduced the risk of listeriosis by 0.35 log. According to Omac et al. [15], sanitising baby fresh spinach either with aerosolised peracetic acid (80 ppm for 20 min) or with ClO_2_ gas (10 ppmv for 20 min) reduced the estimated mean cases of listeriosis by 22%. The same authors determined that combining the three processes, water washing, irradiation and MAP, reduced the mean cases of listeriosis by 65% (Table 2).

Nonetheless, none of the QRA models simulating chemical sanitising took into account that the concentration of the sanitiser, and the time and temperature of exposure of the produce, drive the level of microbial reduction attained by the sanitising treatment [48]. A realistic representation of washing and sanitising in the processing module should disaggregate the effects of sanitising by type of sanitiser, attending to their effectiveness to reduce *L. monocytogenes* depending upon treatment time and temperature, sanitiser concentration and the interaction with the product itself. A sanitisation module could be built upon the mixed-effects meta-regression models estimating *L. monocytogenes* reductions in various types of produce by specific chemical sanitisers presented in Prado-Silva et al. [48].

### 4.3. Cross-Contamination during Processing

In the processing environment, cross-contamination can occur from human carriers, from harbourage sites, and from surfaces in contact with the produce such as conveyor belts, shredders, centrifuges, sorting tables, containers and packaging machines. The survival of *L. monocytogenes* in the processing environments is key to its transmission to foodstuffs. According to Leong et al. [49], *L. monocytogenes* can persist in a processing facility for weeks, and then re-contaminate the product passing through that facility. From the pool of produce QRA models, cross-contamination events were evaluated in five QRA models. In all cases, cross-contamination was modelled by transfer coefficients, which constitute a simple empirical approach that depends heavily on source, recipients and number of contacts [50]. Transfer coefficients during processing stages were defined for packaging and handling steps [14], and for handling mistakes, conveyor belts, and packing equipment [15]. The QRA models did not simulate cross-contamination events during shredding.

In the simulation models, the contribution of cross-contamination during processing to the final risk was variable; a low contribution was found for fresh-cut romaine lettuce [14] and fresh baby spinach [15]. In the former, the occurrence of cross-contamination increased the risk in the susceptible population by 0.06 log (by increasing only in 18% the concentration of *L. monocytogenes* at consumption), whereas, in the latter, the reduction in *L. monocytogenes* attained by sanitising baby spinach was found to counterbalance the increase in *L. monocytogenes* due to cross-contamination, resulting in a reduction in mean listeriosis cases by 12%. On the contrary, Guzel [14], in his QRA model for fresh-cut cantaloupe, estimated that cross-contamination during packaging and handling has a greater contribution to the annual cases of listeriosis (300% increase) than a possible scenario of home temperature abuse of maintaining the product at an ambient temperature for 1 day (220% increase), both measured against a baseline scenario without interventions, cross-contamination and temperature abuse. Nonetheless, the measured effects of cross-contamination on the final risk cannot be directly comparable since they depend on the base assumptions. For instance, it is probable that the low transfer coefficient (0.002) used to model cross-contamination in the QRA model for fresh-cut romaine lettuce [14] has driven the low increase in the final risk, since the occurrence of the cross-contamination leads only to a 0.1 CFU/g increase in the mean concentration of *L. monocytogenes* at the time of consumption in comparison to the baseline scenario.

More complex cross-contamination models, such as discrete-event models [51], have not been applied in any of the QRA models retrieved for produce. Recently, agent-based models of contamination dynamics of *Listeria* spp. for produce packing facilities have been proposed with agents that represent equipment surfaces and employees, each having customised characteristics [52,53]. The agents are assumed to operate autonomously with other agents and the environment, including floors, walls and ceiling. Such models have been proven to be useful for understanding the spread of *Listeria* within the simulated facility, for assessing the efficacy of a variety of sampling plans schemes, and for helping determine corrective actions. However, since these agent-based models are built upon the facility’s design, employees’ shifts, location of employees and equipment, which change between processing facilities, and even in time within a facility, their outcomes are specific to an individual processing facility, allowing an industry to make better decisions on how to detect potential issues quicker and with fewer samples and resources.

The relevance of cross-contamination during the processing stage can be illustrated by a cantaloupe outbreak investigation [54], where subtyping analysis revealed that the processing environment was the main source of contamination. The fact that *L. monocytogenes* forms biofilm on several abiotic surfaces contributes to its persistence and subsequent contamination of post-harvest produce [55]. Another trait of *L. monocytogenes* aiding to persistence in the processing environment is its ability to enter a viable-but-nonculturable (VBNC) state. The cleaning and disinfection procedure leads to a loss in the culturability of *L. monocytogenes* and the appearance of VBNC populations. Subsequently, upon entry into suitable environments (i.e., harbouring site), VBNC *L. monocytogenes* can recover its culturability and begin to proliferate, even remaining infectious [56].

### 4.4. Risk Factors at Retail

The QRA models of a shorter scope (i.e., retail-to-table) start with an estimation of the initial prevalence and/or counts of *L. monocytogenes* in the product, modelled either from data obtained at the end of processing or at the point of retail. Although these listeriosis QRA models commonly tested what-if scenarios reducing the initial prevalence or counts, their authors did not indicate how these reductions could be achieved in the processing stage or earlier in the primary production. For instance, in their retail-to-table model, Franz et al. [16] recognised that the risk of listeriosis from the consumption of leafy greens from salad bars in the Netherlands was heavily driven by the initially assumed prevalence (r = 0.75). In the consumption model of Zoellner et al. [21] for non-RTE frozen vegetables, classification tree analysis highlighted that the initial concentration of *L. monocytogenes* in the lot was the main predictor of illness in the frozen vegetables lot, whereas in the model of EFSA BIOHAZ [22] for non-RTE blanched frozen vegetables, a change in the initial prevalence of *L. monocytogenes* in the product from 9.8% to 13.3% increased the risk by a factor of 1.2. Similarly, for RTE leafy vegetables, Sant’Ana et al. [17] estimated that by reducing the mean initial prevalence of *L. monocytogenes* (at end of processing) from 1.7% to 0.17%, the mean cases of listeriosis in Brazil would decrease by 84%, whereas reducing the maximum initial counts from 2.74 to −1.04 log CFU/g would decrease the mean cases by 91% (Table 2). A discussion on which processes or interventions would be more efficient in helping attain such rather large reductions in prevalence and concentrations was not provided in Sant’Ana et al. [17]. Nevertheless, bringing together the results from the models with processing-to-table scope (Table 2), it could be suggested that highly effective intervention strategies are ionising radiation and cold atmospheric plasma. Milder technologies to control *L. monocytogenes* are sanitisation and MAP; nonetheless, they are more practicable in the produce industry. However, none of the QRA models evaluated the combined effectiveness of sanitisation and MAP (a very likely scenario to occur in industry) to reduce exposure and risk. Moreover, none of the QRA simulations separately modelled the effect of the various sanitisers used in processing. For leafy vegetables processing, chlorine is still widely used for being low cost, and PAA for its activity over a wide pH range and limited reaction with organic matter. For fresh-cut products, there are a growing number of sanitising compounds, including chlorine dioxide, hydrogen peroxide, organic acids, ozone, and electrolysed water, among others [57].

Two QRA models investigated the risk of listeriosis associated with leafy green salads from salad bars in the Netherlands under an end processing-to-table scope, therefore covering inputs that characterise the processing facility for processing fresh-cut vegetables, the distributor and a catering outlet [16,18]. Both studies coincided in that the supply chain of leafy green vegetables for salad bars can be considered as a risk amplifier with respect to *L. monocytogenes*. In Franz et al. [16], to model exposure, actual time–temperature data from the processing plant to restaurant were acquired as well as time–temperature trajectories of the vegetables kept in the catering establishment. This model showed that, despite the fact that maintaining the temperature controlled at 2–5 °C from the processing plant to the restaurant, *L. monocytogenes* in leafy greens could grow from a mean of 250 CFU/g at the end of processing to 735 CFU/g (95% CI: 690–780 CFU/g) at consumption because, in the salad bar, temperatures can fluctuate up to 13 °C. This resulted in a mean dose per contaminated portion of 85,000 CFU, which increased to 105,000 CFU (95% CI: 28,500 to 251,000 CFU) in a hypothetical scenario of a breakdown in the salad bar’s cooling unit that increased the temperature up to 18 °C. Despite the temperature fluctuations having a direct impact on the contaminated servings, the risk of infection from any serving of leafy green salads from salad bars was driven mainly by the initial prevalence of *L. monocytogenes* at retail (r = 0.75) and the portion size (r = 0.62). Tromp et al. [18] refined the model of the leafy greens salads from salad bars, (1) by introducing the notion that in reality storage times are interdependent, meaning that if a product remains for a long time in one step, it will remain a shorter time in a following step due to less remaining shelf-life; and (2) by modelling logistics in order to incorporate logistics performance indicators such as shrinkage (product loss) and out-of-stock. Tromp et al. [18] found that modelling logistics—as opposed to Franz et al. [16]—increased the estimated risk of listeriosis 4-fold. In addition, from a logistics point of view, increasing the delivery frequency in the catering establishment from 2 to 5 times per week halved the mean number of cases of listeriosis in the general population from 1.43 to 0.70 cases per annum. Through the results of this model, it was clear that having shorter storage times with greater delivery frequency resulted in reduced growth of *L. monocytogenes*.

Time–temperature measurements indicate that the cold storage in display cabinets at retail is generally not the most efficient step in the cold chain, and that temperatures frequently rises above the recommended limit [58,59]. In any case, at retail or in salad bars, temperature abuse has been often pointed out as a key factor contributing to increasing the risk of listeriosis. Therefore, in a QRA model, it is very important to be able to capture the growth of *L. monocytogenes* during distribution and retail as realistically as possible by representative time–temperature profiles as well as accurate kinetic data and validated growth models of *L. monocytogenes* in the specific product.

### 4.5. Risk Factors at Home

At the consumer level, temperature abuse has been tested by some of the QRA models, whereas in Carrasco et al. [12], the home storage temperature had a stronger effect on the number of listeriosis cases associated with RTE lettuce salad than home storage time and *L. monocytogenes* concentration at consumption; Guzel [14] predicted that keeping the fresh-cut romaine lettuce or the fresh-cut cantaloupe for 24 h at 20 °C (ambient temperature) would increase *L. monocytogenes* populations at consumption by 56% or 300%, respectively, with the consequent increase in risk of listeriosis in the susceptible population by 1.1 log or 2.95 log, respectively. Nonetheless, keeping the fresh produce for one day on the countertop is an unusual consumer practice. In a more credible scenario, where fresh spinach was left at ambient temperature for 1.2 h, Omac et al. [15] estimated that this practice would increase the mean cases of listeriosis by 55% (no temperature abuse in the baseline scenario). The only QRA model where domestic temperature did not determine the exposure to *L. monocytogenes* was that of Zoellner et al. [21] for non-RTE frozen vegetables, where in sensitivity analysis neither the time stored at room temperature (Pearson’s coefficient of correlation r = 0.02) nor the time or temperature in the refrigerator (r = 0.01) correlated with the dose of *L. monocytogenes* per serving. Furthermore, they found that the consumer practice of thawing frozen vegetables did not affect the risk of illness. Domenech et al. [20] assessed the effects of other habitual consumer practices, and found that the probability of washing (r = −0.46–−0.43) followed by the time under running tap water (r = −0.09–−0.14) were parameters that could reduce the prevalence of *L. monocytogenes* the most in fresh lettuce at the domestic level (Table 2).

Fresh and RTE minimally processed produce are products that have a very short shelf-life because of their high perishability caused by their high moisture content; in particular, their wounding caused by processing leads to many physical and physiological changes that noticeably affect their quality [60]. Notwithstanding these facts, some QRA models have assessed unrealistic what-if scenarios of extending consumption times, probably to make the point that home storage time is determinant of the risk of listeriosis. For instance, the models of Guzel [14] estimated that increasing the consumption time of fresh-cut romaine lettuce and fresh-cut cantaloupe to a maximum of 14 and 10 days, increased the mean *L. monocytogenes* concentration (CFU/g) at consumption by 2100% and 2300%, respectively. On the contrary, according to Carrasco et al. [12], reducing the shelf-life of RTE salad from a maximum of 12 days to a maximum of 4 days decreased by 84% the annual cases of listeriosis (Table 2).

The two QRA models consisting of a sole consumption module were conducted for non-RTE frozen vegetables [21,22]. The QRA model of Zoellner et al. [21] was prompted by the multi-country outbreak of listeriosis linked to frozen corn that caused 53 cases and 10 deaths over the period of 2015–2018. Using whole-genome-sequencing (WGS), the outbreak investigation concluded that the environmental contamination of a frozen vegetable plant in Hungary was the source of the strain, which persisted despite cleaning and disinfection [61]. It was highlighted that some frozen fruits and vegetables can be added uncooked to salads or used in smoothies or other products without being subjected to any process to eliminate or reduce the level of pathogens [62]. For that reason, the QRA model of Zoellner et al. [21] focused only on the consumption module, aiming to understand to what extent consumer preparation methods different from the packaging instructions impact on the risk of listeriosis. A similar objective was pursued by the subsequent QRA model of EFSA BIOHAZ [22] for blanched frozen vegetables. Zoellner et al. [21] found that cooking the vegetables before serving (r = −0.87) and log reduction due to proper cooking (r = −0.48) were the most important drivers for reducing the dose of *L. monocytogenes* consumed; when at least half the servings are properly cooked, the number of illnesses per consumed lot is 0 (95% CI: 0–1) in the susceptible population. Likewise, the QRA model of EFSA BIOHAZ [22] estimated that reducing the proportion of uncooked servings from 23% (baseline) to 4% (best case scenario) decreases the predicted listeriosis cases per year from 1.62 to 0.041 in elderly females. Furthermore, they estimated that cooking the frozen vegetables reduces the listeriosis cases per 10^12^ servings from 400 to 0.23 in the female elderly population, and from 1900 to 0.53 listeriosis cases in the male elderly population (Table 2).

Serving size was evaluated in Carrasco et al. [12] and Franz et al. [16] for RTE lettuce and salad bar leafy greens, respectively, and in both cases, it had a moderate effect on the risk of infection. In the model of EFSA BIOHAZ [22], a change in serving size from 49 to 106 g was found to increase the probability of illness per serving of uncooked frozen vegetables by a factor of 2.2. In contrary, in the model for frozen vegetables by Zoellner et al. [21], serving size did not drive the final risk of listeriosis. This difference arises from the fact that frozen vegetables are mostly eaten cooked, which attenuates the effect of increasing the serving size.

### 4.6. Cross-Contamination during Consumer’s Handling

The cross-contamination during the consumer’s handling was modelled using transfer rates for contaminated boards, hands and knives [20], and for unwashed cutting boards or countertops, kitchen tools, and unwashed hands [11]. From the QRA models retrieved, the relative importance of cross-contamination during the consumer’s handling was assessed only by Domenech et al. [20]. They found that the parameters related to cross-contamination such as the contamination of surfaces (r = 0.23–0.29) and chopping board and knife transfer rates (r = 0.07–0.13) were the parameters that increased the consumers’ exposure to *L. monocytogenes* the most, in comparison to contamination at retail (r = 0.02–0.04), and refrigeration temperature and storage time (r = 0). These assessments highlighted the importance of accurately representing the transfer coefficients in the cross-contamination modules, since they have been shown to moderately drive the final risk estimate of listeriosis.

### 4.7. Availability of QRA Scripts

Appendix A provides information about model availability. Most of the models were developed using @risk software. Only the EFSA model (R model) is directly accessible [63]. Zoellner et al. [21] indicate that the models will be made available on request to the authors. This lack of availability of the models raises questions about the ability of other risk assessors to construct new developments. Each time, it seems necessary to start from scratch without being able to rely easily on existing models. The more systematic sharing of spreadsheets and scripts would benefit the entire community.

## 5. Conclusions

It would be advantageous that a QRA model for produce includes a primary production module so that the contributions to risk from important on-farm factors such as type of cultivation, water, fertilisers and irrigation method could be introduced. This would enable the evaluation of current on-farm practices, control measures and the potential application of more stringent standards of production to reduce the exposure to *L. monocytogenes*. Despite the availability of data on the effectiveness of the use of sanitisers to reduce the load of *L. monocytogenes* in produce, none of the QRA modules contained a module for washing that could estimate microbial reduction as a function of the type of sanitiser, its concentration and the time of exposure. A realistic QRA model for minimally processed produce should take into account such an informative module. Furthermore, since secondary contamination can occur in the processing plants from equipment and environmental elements, the most relevant stages and occasions for cross-contamination during processing should be pinpointed and modelled. This would enable the assessment of control measures that can limit the occurrence of cross-contamination events, such as the implementation of more stringent controls for raw materials, environmental monitoring programs and/or sanitation procedures. Since the safety of produce, in particular for RTE products, heavily relies on the maintenance of low temperatures in the supply chain, *L. monocytogenes* growth should be simulated using representative data of time and temperature alongside distribution, retail and home storage, as well as accurate microbial kinetic parameters for the specific foodstuff.

## Figures and Tables

**Table 1 foods-13-01111-t001:** Features of quantitative risk assessment models of *L. monocytogenes* (LM) from consumption of produce by scope.

Scope	Food	RTE	Cross-Contamination	DR–End-Point	Type of DR Model	DR Sub-Populations	Strain Variability	Temp Profiles/ Lagtime	Country	Source
Farm-to-table	Lettuce	No	Yes: Transport, market, restaurants and at home	Exp–I	FAO/WHO [23]	High-risk/Low-risk	LM strain diversity implicit in r	No/Yes	Korea	Ding et al. [11]
Processing-to-table	RTE lettuce salad	Yes	No	WG–I	Farber et al. [24]	High-risk/Low-risk	-	No/No	Spain	Carrasco et al. [12]
	RTE lettuce salad	Yes	No	Exp–I	FAO/WHO [23]	High-risk/Low-risk	LM strain diversity implicit in r	Yes/No	France	Crèpet [13]
	Fresh-cut romaine lettuce	Yes	Yes: Processing—during packaging	WG–I	Farber et al. [24]	High-risk/Low-risk	-	No/Yes	USA	Guzel [14]
	Fresh-cut cantaloupe	Yes	Yes: Processing—after cutting	WG–I	Farber et al. [24]	High-risk/Low-risk	-	No/Yes	USA	Guzel [14]
	Fresh baby spinach	No	Yes: Processing—before packaging	Exp–I	Chen et al. [25]	General	-	No/Yes	USA	Omac et al. [15]
End processing-to-table	Leafy green salads from salad bars	Yes	No	Exp–I	Chen et al. [25]	General	-	Yes/No	Netherlands	Franz et al. [16]
	RTE leafy vegetables	Yes	No	Exp–I	Buchanan et al. [26]	General	-	No/No	Brazil	Sant’Ana et al. [17]
	Leafy green salads from salad bars	Yes	No	Exp–I Log–D	Chen et al. [25]; Williams et al. [27]	General Perinatal	-	Yes/No	Netherlands	Tromp et al. [18]
Retail-to-table	Fruits and vegetables	Yes	No	Mouse Epi–I	FDA-FSIS [19]	Multiple	Variability in the virulence of different LM strains represented in DR	No/No	USA	FDA-FSIS [19]
	Lettuce	No	Yes: Handling at home	Exp–I	FAO/WHO [23]	High-risk	Strain diversity implicit in r	No/No	Spain	Domenech et al. [20]
Consumption	Non-RTE frozen vegetables	No	No	Exp/WG–I	Buchanan et al. [26] Bemrah et al. [28]	High-risk/Low-risk	-	No/Yes	USA	Zoellner et al. [21]
	Blanched frozen vegetables	No	No	Exp–I	EFSA BIOHAZ [2] based on Pouillot et al. [29]	Elderly population (male, female)	Distribution for EGR 5 °C modelled from LM growth data in corn, green peas, carrots, broccoli, beans and asparagus. LM strain virulence and host susceptibility explicit in r distribution	No/No	EU	EFSA BIOHAZ [22]

DR: dose–response; Exp: exponential model; WG: Weibull-gamma model; Log: logistic model; Mouse-Epi: mouse-epidemiological model; I: illness endpoint; D: death endpoint.

**Table 2 foods-13-01111-t002:** Predictive microbiology models and main outcomes related to what-if scenarios and sensitivity analysis from quantitative risk assessment models of *L. monocytogenes* (LM) from consumption of produce.

Scope	Food	Predictive Microbiology Models	What-If Scenarios	Sensitivity Analysis ^1^	Model Complexity ^2^	Source
Farm-to-table	Lettuce	Growth (Gompertz model, polynomial model for lag phase, growth square root model)	NA	NA	Low	Ding et al. [11]
Processing-to-table	RTE lettuce salad	Growth (linear model, growth square root model)	(1) Use of MAP (5.5%, CO_2_, 3% O_2_; 92.5% N_2_) as opposed to no packaging (baseline) reduce mean number of listeriosis cases by 95%; (2) Reducing the shelf-life from a maximum of 12 days to 4 days reduces number of cases by 84%; (3) Preventing high-risk consumers from consuming RTE salads reduces number of cases by 75%; (4) Applying microbiological criterion at primary production (*n* = 20; c = 0; absence in 25 g) reduces cases by 43%.	Outcome—number of listeriosis cases, ranked in this order: serving size, storage temperature at home, storage time at home, LM concentration at consumption (no r provided)	Medium: An approximation is given to solve growth for dynamic temperatures	Carrasco et al. [12]
	RTE lettuce salad	Growth (logistic model with delay and rupture, cardinal parameter model for temperature). Three models were proposed for the maximum levels of contamination (one based on observed challenge tests, the two others based on the initial contamination in the pack). A model was established to consider lag phase. The models are fully described in [30].	The model enables the assessment of the effect of water chlorination during the washing of lettuce.	The impact of different hypothesis for the risk characterisation was carried. It includes the way of modelling maximum level (y_max_), the consideration of lag, and the clustering of contamination in packed salad (b parameter). The non-treatment or water with chlorine multiplied the risk of listeriosis by 2.	High: Second-order Monte Carlo simulation is used to assess uncertainty of risk of listeriosis. The model starts from the lettuce and takes into account the effect of washing. The model reproduces the cold chain itinerary of the lettuce.	Crèpet [13]
	Fresh-cut romaine lettuce	Growth (Baranyi model, growth square root model)	(1) LM counts at consumption is reduced by >99% after exposure to ionising radiation (1 kGy at room temperature) and reduces log risk of illness by 1.66 log in the susceptible population; (2) Cold atmospheric plasma reduces LM population by 92% and log risk in 1.34 log; (3) Peroxyacetic acid reduces LM counts by 28%, and log risk by 0.35 log; (4) Cross-contamination during processing increases LM counts in 18% and log risk by 0.06 (because the transfer coefficient was very low at 0.002); (5) Home temperature abuse (20 °C × 24 h) increases mean LM counts by 56% and log risk in 1.1; (6) Consumption time up to a maximum of 14 days increases LM counts by 2100% and log risk in 2.6.		Low	Guzel [14]
	Fresh-cut cantaloupe	Growth (Baranyi model, growth square root model)	(1) Implementation of irradiation reduces LM at consumption by 99.9%, and mean cases of listeriosis by 99%; (2) Cross-contamination during processing increases cases by 300%; (3) Home temperature abuse at home (20 °C × 24 h) increases LM at consumption by 300% and cases by 220%; (4) Extending consumption time up to a maximum of 10 days increases LM at consumption by 2300% and cases in 39,000%.	NA	Low	Guzel [14]
	Fresh baby spinach	Growth (Baranyi model, square root models for growth and lag phase, polynomial model for y_max_,)	Baseline scenario represents neither interventions during processing nor cross-contamination. (1) Washing with water decreases mean cases of listeriosis by 7.5%; (2) Water with PAA or ClO2 reduces mean cases by 22%; (3) Washing and cross-contamination still reduces mean cases by 12%; (4) Washing plus temperature abuse (at home, ambient temperature for 1.2 h) increases mean cases by 55%; (5) Washing plus irradiation reduce cases by 56%; (6) Washing plus irradiation plus MAP reduce cases by 65%; (7) Washing plus cross-contamination plus irradiation plus MAP plus temperature abuse reduce mean cases by 35%.	NA	Medium: Various scenarios tested by sanitiser and combinations	Omac et al. [15]
End process-to-table	Leafy green salads from salad bars	Growth (Baranyi, growth square root model)	(1) A breakdown in the salad bar’s cooling unit (temperature of 18 °C from the moment that the salad bar is filled) increases the mean number of cases in 23% (In the baseline scenario temperature of salad bar is assumed to fluctuate normally between 0 and 13 °C).	Outcome—probability of infection from any serving: Initial prevalence (r = 0.75) and portion size (r = 0.62).	Medium: temperature profiles	Franz et al. [16]
	RTE leafy vegetables	Growth (linear model, growth square root model)	(1) Reducing mean initial prevalence of LM from 1.7% to 0.17% decreases mean cases of listeriosis by 84%; (2) Reducing initial mean prevalence and keeping temperature strictly between 1 and 5 °C along processing and storage reduces cases by 85%; (3) Reducing maximum initial counts of LM from 2.74 to −1.04 log CFU/g reduces mean cases by 91%; (4) Reducing maximum initial counts and keeping temperature strictly between 1 and 5 °C reduce cases by 92%; (5) Reducing both prevalence and counts decreases mean cases by 98.7%.	NA	Low	Sant’Ana et al. [17]
	Leafy green salads from salad bars	Growth (linear model, growth square root model)	(1) The delivery frequency towards the restaurant was increased from 2 days a week to 5 days a week. In this scenario, the catering outlet is allowed to order leafy green–based salad products every weekday. This scenario halved the mean number of cases.	Outcome—the desired service level with regard to “out-of-stock” (z parameter; the greater z is, the smaller the	Medium: temperature profiles	Tromp et al. [18]
Retail-to-table	Fruits and vegetables	Growth (linear model, square root model for EGR)	NA	NA	Medium: Fruits and vegs considered in separate; dose–response models developed for three subpopulations	FDA-FSIS [19]
	Lettuce	Growth (linear at 6 °C and 23 °C), Survival (empirical equation for water treatment)	NA	Outcome—LM counts at consumption: probability of washing (r = −0.46–−0.43), surface contamination (r = 0.23–0.29), time under running tap water (r = −0.09–−0.14), board/knife transfer rate (r = 0.07–0.13), contamination at retail (r = 0.02–0.04)	Low	Domenech et al. [20]
Consump-tion	Non-RTE frozen vegetables	Growth (linear, EGR square root model, empirical model for lag phase)	The median log risk of listeriosis from consumption of frozen vegetables contaminated with LM is −12.7. (1) Within-package clustering parameter between 0.01 and 0.1 in the baseline—as opposed to 1 in the baseline—reduces median log risk to −15/−14.1; (2) Number of packages tested per lot of 20 or 10—as opposed to 5 in the baseline—reduces median log risk to −14.4/−13.7; (3) Thawing at ambient temperature or in the fridge has negligible effect on the risk; (4) Changing the number of servings per meal (s = 0.5, 2) also resulted in no difference from the baseline (s = 1) risk of listeriosis.	Outcome—dose of LM consumed: cooking the serving (r = −0.87), log reduction due to proper cooking (r = −0.48), LM counts in a serving from contaminated package (r = 0.46), time stored at room temperature (r = 0.02), time/temperature in the refrigerator (r = 0.01)	Medium: Bacterial clustering in a package is represented; partitioning of the package into portions is modelled; handling prior to consumption such as cooking and thawing is included	Zoellner et al. [21]
	Blanched frozen vegetables	Growth (linear, square-root model using EGR 5 °C)	(1) In elderly females, cooking the vegetables reduces the risk of listeriosis per serving by 3.2 log (from −9.4 to −12.6 log), and the number of cases per 10^12^ servings from 400 to 0.23; (2) In elderly males, cooking the vegetables reduces the risk of listeriosis per serving by 3.6 log (from −8.7 to −12.3 log), and the number of cases per 10^12^ servings from 1900 to 0.53; (3) Reducing the proportion of uncooked servings from 23% to 4% reduces the predicted listeriosis cases per year from 1.62 to 0.041 in elderly females.	Outcome—Probability of illness per serving from uncooked frozen vegetables: MPD from 7.8 to 9.8 log CFU/g increases risk by 2.5; serving size from 49 g to 106 g increases risk by 2.2; initial prevalence from 9.8% to 13.3% increases risk by 1.2.	Low: Generic model; only demands some knowledge in R software to utilise it	EFSA BIOHAZ [22]

^1^ The outcome variable on which sensitivity analysis is carried out is indicated; “r” refers to the Pearson’s correlation coefficient. ^2^ Model complexity was assessed by the authors in terms of easiness of reproducibility. When considered as of medium or high complexity, explanations are provided. aw: water activity; LPD: lag phase duration; RLT: relative lag time; GR: maximum growth rate; EGR_x_: exponential growth rate at x °C; LAB: lactic acid bacteria; LAC: lactic acid concentration; RR: risk reduction; NA: not addressed in the study.

## Data Availability

Data is contained within the article. The BibTeX file containing records used in this systematic review is available.

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
