# Peer review of "A Critical Review of Risk Assessment Models for Listeria monocytogenes in Produce"

_foods, 2024, doi:10.3390/foods13071111_

Round 1

Reviewer 1 Report

Comments and Suggestions for Authors

The authors present an interesting comparative study of microbial risk assessment models for fresh produce. It is generally well-written, contains useful information, and would constitute a useful addition to the body of scientific literature on the subject. The following are the recommended changes and minor additions.

The discussion and conclusion could benefit from three things. 1) A clearer summary of the factors that, according to the models, have the biggest influence on the likelyhood of illness in the consumers. 2) A brief comment on the practicality of the measures suggested by the models, e.g.ionizing radiation has consumer acceptibility issues, increasing the frequency of deliveries has an economic cost etc. 3) In cases where the models do not agree with each other (e.g. lines 434-437) this should be investigated a little more deeply if possible. Why do they give different results?

Language corrections:

Line 49 “vegetables’ groceries” – incorrect and unclear, does it mean “retail vegetables”?

Line 178 Challenging to control, not “to be controlled”

Line 196 delete “then”

Line 201, 208, 402 comma not semicolon

Line 221 ...far more effective…

Line 231, 243, 244, 322, 323, 382, 408 reduce/decrease/increase by, not in.

Line 275 delete “even”

Line 276 delete “still”

Line 276 ...due to cross-contamination….

Line 349 up to 13°C

Line 339 delete semicolon

Line 352 Despite the fact that

Lines 381-382 ...at consumption by 56% and 300% respectively.

Other corrections, suggested changes, and queries:

Lines 66 & 77 contradictory: The method says the literature search used 2000 as a starting date, but the results mention 1998 as the oldest model.

Lines 93-96. Why is the utility of study 11 not clear?

Table 1: “WG” not defined in footnote. “NA” is defined in the footnote but does not occur in the table.

Table 2 has lots of information but is extremely cumbersome. The authors should consider some or all of the following. 1) Split the table, e.g. have the What-if scenarios seperately. 2) Move some content from the table to the main text where possible. 3) Write more concisely – in a table, it is not always necessary to use complete sentences, text can be in note form.

Line 112. Change “...choice in all models…” to “…choice of secondary model in all cases…”

Lines 119-120 Maximum levels higher in challenge tests. This is quite an interesting observation that has implications for a lot of models. Perhaps it should be expanded upon.

Lines 268-269. The models did not include cross-contamination during shredding, but is this not likely to be quite important? Is there a reason none of the models include it?

Line 270-282 and 296-305. From the information presented here it seems that cross-contamination during processing is a major risk factor for canteloupe but not for leafy vegetables. The authors should expand on this a little and comment on the reasons for the difference.

Lines 326-337 seem more appropriate in section 4.2

Comments on the Quality of English Language

The English language used is generally good. A few minor corrections are required, which have been listed for the authors' attention.

Author Response

see file enclosed. thanks.

Reviewer 2 Report

Comments and Suggestions for Authors

This review carried out quantitative risk assessment (QRA) models of L. monocytogenes in produce with the objective of appraising and contrasting the effectiveness of the control strategies placed along the food chains. This is a series of reviews with good logic and tight organization for quantitative risk assessment of Listeria monocytogenes. The whole content is also somewhat profound, connecting the preceding and the following, and providing a good reference and summary for subsequent researchers. It is recommended to focus more on the interpretation and discussion of the information in the table.

Some detail: 

Since L90, think over the food chain expression, what about to change into from production to consumption, better than farm to fork. 

Tab 2 title used LM, but not in Tab 1 title? Check the first spelling for LM 

L107 not the “predictive microbiology”, suggest to “predictive microbiological models” 

L117 check this “maximum levels” and following, do you mean “maximum bacterial levels”? 

L118 Listeria monocytogenes to L. monocytogenes

L190 “Now dose-response relationships based on strain genetics have been proposed” suggest to add some words, like ST information. 

4.7 about the title, not the models, suggest to like “software or platform” 

Suggest to shorten the conclusion, now is too long, and some detail might be combined in the above parts.

Author Response

see file enclosed. thanks.

Reviewer 3 Report

Comments and Suggestions for Authors

The manuscript comprehensively reviews the existing QRA models for Listeria monocytogenes in produce. The authors have published a serial of review paper related to listeria monocytogenes QMA models in various foods. The manuscript provides insightful content, and with some expansions, it can contribute significantly to the discussion on improving QRA models in the food industry.

One point may help to enrich the information. The paper mentions that most QRAs are focused on fresh or ready-to-eat leafy greens, it would be advantageous to expand upon this, possibly investigating the lack of models for other types of produce, and the risk in what other type of produce should be concerned, and indicate more study on the predictive models and QRA models need to be done.

Author Response

see file enclosed. thanks.

Reviewer 4 Report

Comments and Suggestions for Authors

The analysis of quantitative risk models has the practical advantage to evaluate the effectiveness of control strategies along the food chains. The indication of future QRA models containing a sanitisation module seems particularly wothwhile and could be better described in Discussion, giving further value to Your very interesting and accurate work

Author Response

see file enclosed. thanks.
